# Research on a Novel Unsupervised-Learning-Based Pipeline Leak Detection Method Based on Temporal Kolmogorov–Arnold Network with Autoencoder Integration

**DOI:** 10.3390/s25020384

**Published:** 2025-01-10

**Authors:** Hengyu Wu, Zhu Jiang, Xiang Zhang, Jian Cheng

**Affiliations:** 1College of Energy and Power Engineering, Xihua University, Chengdu 610039, China; 212023080700018@stu.xhu.edu.cn (H.W.); cj@stu.xhu.edu.cn (J.C.); 2Key Laboratory of Fluid and Power Machinery, Xihua University, Ministry of Education, Chengdu 610039, China

**Keywords:** pipeline leak detection, temporal Kolmogorov–Arnold network (TKAN), autoencoder (AE), unsupervised learning, time series anomaly detection

## Abstract

Artificial intelligence (AI) technologies have been widely applied to the automated detection of pipeline leaks. However, traditional AI methods still face significant challenges in effectively detecting the complete leak process. Furthermore, the deployment cost of such models has increased substantially due to the use of GPU-trained neural networks in recent years. In this study, we propose a novel leak detector, which includes a new model and a sequence labeling method that integrates prior knowledge with traditional reconstruction error theory. The proposed model combines the Kolmogorov–Arnold Network (KAN) with an autoencoder (AE). This model combines the Kolmogorov–Arnold Network (KAN) with an autoencoder (AE), forming a hybrid framework that effectively captures complex temporal dependencies in the data while exhibiting strong pattern modeling and reconstruction capabilities. To improve leak detection, we developed a novel unsupervised anomaly sequence labeling method based on traditional reconstruction error theory, which incorporates an in-depth analysis of the reconstruction error curve along with prior knowledge. This method significantly enhances the interpretability and accuracy of the detection process. Field experiments were conducted on real urban water supply pipelines, and a benchmark dataset was established to evaluate the proposed model and method against commonly used models and methods. The experimental results demonstrate that the proposed model and method achieved a high segment-wise precision of 93.1%. Overall, this study presents a transparent and robust solution for automated pipeline leak detection, facilitating the large-scale, cost-effective development of digital twin systems for urban pipeline leak emergency management.

## 1. Introduction

China has abundant water resources as a whole, but the per capita water resources are only a quarter of the global average, and their distribution is extremely uneven [1]. Due to long operating years, installation issues, and environmental factors, water supply pipeline leaks frequently occur, posing significant challenges to the efficiency and safety of urban water supply systems. The average leakage rate stands at 13.4%, substantially higher than international standards, as detailed in the “China Urban and Rural Construction Statistical Yearbook” [2,3]. This not only results in water wastage but also raises the risk of secondary pollution. Therefore, enhancing the maintenance and management of water supply systems to reduce leakage is crucial for ensuring the security and sustainable use of water resources.

With the continuous deepening of research work on this issue, scholars have achieved a large number of results. Methods based on fluid dynamics models and finite element simulation have been developed by using advanced mathematical models to reveal complex mechanics principles, such as model-based residual generation with friction identification approaches [4], CFD-FEA based approaches [5], Kalman filter approaches [6], lateral extraction hydraulic gradient approaches [7,8,9,10], Fisher’s discriminant analysis approaches [11], and so on. The effectiveness of these leakage detection methods across various pipeline networks has also been broadly recognized. In addition, data-driven approaches provide valuable alternatives that function like a “black box” and can effectively capture and learn the intricate relationships between fluid flow signals in pipelines without the need for a detailed understanding of underlying physical principles and mechanisms.

Artificial intelligence (AI) technology, recognized for its efficiency and reliability, has been widely applied across various fields [12], and its utility in the domain of water supply pipeline leak detection has demonstrated promising outcomes. Notably, Quinones-Grueiro et al. [13] illustrated the feasibility of Principal Component Analysis (PCA) as an unsupervised learning technique for this purpose, highlighting its practical application in identifying anomalies. In the same year, Quinones-Grueiro et al. [14] conducted a comparative study of different supervised learning methods, concluding that Support Vector Machines (SVMs) exhibit superior performance and robustness in leak detection tasks. Yingchun Xie et al. [15] developed a convolutional neural network (CNN) model based on time–frequency distribution maps. Zhang Peng et al. [16] proposed a new method for water pipeline leak detection based on a Pseudo-Siamese Convolutional Neural Network (PCNN). Additionally, autoencoders are a type of neural network that can be trained on unlabeled data and can distinguish attributes that may deviate from the normal state. This feature is very useful for pipeline fault detection [17,18]. In addition, autoencoders (AEs) are often integrated with other models in anomaly detection tasks to form hybrid approaches, as their characteristics can effectively address the limitations of other models, particularly in terms of feature extraction and pattern modeling. The reconstruction properties of AEs align closely with the theory of reconstruction error, enabling them to deliver effective anomaly detection capabilities [19,20,21,22,23]. Additionally, AEs complement other models, significantly enhancing detection performance. Moreover, AEs exhibit inherent noise resistance, which ensures robust performance in industrial-scale real-world datasets. This characteristic eliminates the need for additional data denoising procedures, thereby enabling real-time monitoring [24,25,26].

For pipeline leak detection tasks, negative pressure waves have been proven to be the most effective and widely used method [27]. Because the negative pressure wave signals used in pipeline leak detection are a type of time series signal, recurrent neural networks (RNNs) offer significant advantages compared to other models [28,29]. In addition, Long Short-Term Memory (LSTM) has been widely applied in pipeline leak detection tasks. Its unique gating mechanism effectively captures both long-term and short-term dependencies in time series data, significantly enhancing the model’s ability to predict and understand dynamic changes in sequences [30]. Kim et al. [31] proposed a method called LSTM-RNN, which can improve the accuracy and efficiency of leak source detection. Lei Yang et al. [32] proposed a method combining OPELM and bidirectional LSTM, which can accurately detect leak events and significantly reduce the number of false alarms compared to existing methods. However, existing deep learning models have undergone a debatable development process, characterized by ever-increasing model sizes, deeper layers, and more network parameters. Nowadays, training a model often requires hundreds or even thousands of GPU days, significantly increasing the cost of AI usage and reducing its capability for widespread application in everyday life [33]. Recently, Ziming Liu et al. [34] proposed a groundbreaking new neural network architecture—the Kolmogorov–Arnold Network (KAN). Due to its high accuracy, strong interpretability, and fewer model parameters, KAN has caused an unprecedented sensation and is considered a potential alternative to the cornerstone of modern deep learning, Multilayer Perceptrons (MLPs). To address the limitations of traditional models in handling complex sequential patterns, Rémi Genet et al. [35] proposed the Temporal Kolmogorov–Arnold Network (TKAN)—an innovative neural network architecture that integrates LSTM with the KAN network. This innovation enables more accurate and efficient multi-step time series predictions. The TKAN architecture has shown significant potential in fields requiring multi-step forecasting. However, to the best of the authors knowledge, the application of TKAN to real-time automated pipeline leakage detection has not been explored yet.

This study aims to propose an unsupervised automated pipeline leakage detector by integrating the Temporal Kolmogorov–Arnold Network (TKAN) with an autoencoder (AE) to develop a hybrid model called TKAN–autoencoder (TKAN-AE). Additionally, a novel anomaly sequence labeling method based on reconstruction error, combined with prior knowledge, is proposed to enhance the reliability and accuracy of automated leakage detection. On-site experiments were conducted in real pipelines with simulated leaks in megacities such as Shanghai to build a benchmark dataset. The detector was trained on unlabeled data and compared against two widely used baseline time series models as well as classical threshold-based labeling methods. The major contributions and innovations of this study are as follows:

(1) This paper presents a novel hybrid model called TKAN-AE, where the Temporal Kolmogorov–Arnold Network (TKAN) is employed to capture complex temporal dependencies within the data, while an autoencoder (AE) is used for compressing and reconstructing the input data. The model demonstrates well-balanced performance in pipeline leak detection tasks, achieving an F1 score of 75%.

(2) Building on the traditional reconstruction error theory, we incorporate prior knowledge to deeply analyze the reconstruction error curve and the negative pressure wave characteristics of pipeline leaks. We propose a new anomaly sequence labeling method. Compared to traditional methods, this approach improves the segment-level precision by 54.2% and was experimentally validated to be applicable to real-world field datasets and various models, demonstrating strong robustness and generalizability.

(3) To the best of our knowledge, this is the first time that the lightweight TKAN network has been applied to real-world field datasets for urban water pipeline leak detection. This work provides a feasible solution for the future industrial-scale application of artificial intelligence in pipeline leak detection in real-world scenarios.

## 2. The Proposed Leak Detection Method

The whole structure of the proposed leak detector in this paper is illustrated in Figure 1, including data preprocessing, the construction of TKAN-AE, and anomaly labeling. Initially, the negative pressure wave data, which are obtained from real city pipelines, undergoes preprocessing steps like basic noise reduction, applying sliding window averages, and normalization. This results in turning the original data into subsequence datasets. Next, the subsequences are compressed by the encoder into latent variables and then reconstructed by the decoder. Afterward, the reconstructed data are compared with the original dataset, and the reconstruction loss for each individual point is measured. Next, the reconstruction error curve is plotted, and our proposed anomaly sequence labeling method is applied to detect anomalies.

### 2.1. Data Preprocessing

Data preprocessing, including cleaning and denoising, is considered one of the important parts in a machine learning process. It has been shown by Y Zhang et al. [36] that directly using data that is noisy or not clean for training neural networks may not be an ideal practice. Proper preprocessing can improve the training effectiveness and predictive performance of the model. The first step involves using the sliding window average to further reduce random fluctuations in the signal, as per Shtayat et al. [37], thereby enhancing its smoothness. The sliding window average method can express as:(1)$$MA_{t}=\frac{1}{n}\sum_{i=t-n+1}^{t}x_{i}$$
where $MA_{t}$ is the moving average at time $t,x_{i}$ is the value of the time series at time i, and n is the window size (number of periods to average over).

Next, these subsequences from the time series go through sliding window reshaping, which results in a matrix. This matrix then becomes the final form of the input data for the model.

After that, the second step is applying MinMax scaling normalization on the data. Because each data point may belong to a different pressure range, it becomes important to make sure that all training data remain within the same scale [38]. This is necessary to avoid having specific features with larger values affect the anomaly detection model more than others because of their higher magnitudes [39].

### 2.2. TKAN-AE

#### 2.2.1. Autoencoder

Autoencoders perform anomaly detection by learning effective representations of data. Their encoder–decoder architecture adapts well to capturing the underlying structure of data and identifying anomalies. They are a type of neural network used for efficiently reconstructing unlabeled data, learning representations of a given dataset during training to filter out irrelevant aspects like noise [40,41]. In anomaly detection, the model learns patterns of normal processes, classifying any data that deviate from this pattern as anomalous [42].

Autoencoders consist of two main parts, as illustrated in Figure 2.

The first part is the encoder, which maps the input to a latent representation space h, and the decoder maps information from this latent space back to a reconstructed input.

In the simplest architecture with a single hidden layer, the encoder of an autoencoder takes the input $x\in R^{d}$ and maps it to the latent space $h\in R^{p}$. Typically, the latent space h is represented as:(2)$$h=\sigma \left(Wx+b\right)$$
where σ is an activation function like sigmoid or ReLU, W is a weight matrix, and b is a bias vector, usually initialized randomly and updated gradually during training.

Next, the decoder takes the latent representation h and attempts to reconstruct the input of the encoder. In other words, the decoder aims to map the latent representation h back to a reconstructed input $x^{\prime }$. Following previous notation conventions, this operation can be expressed as:(3)$$x^{\prime }=\sigma^{\prime }\left(W^{\prime }h+b^{\prime }\right)$$
where $\sigma^{\prime }$ is also the activation function, and $W^{'}$ and $b^{'}$ are the weight matrix and bias vector specific to the decoder stage.

Ultimately, during the training process, the autoencoder minimizes the loss function, a process known as reconstruction error. Typically, the Mean Absolute Error (MAE) exhibits strong robustness against outliers and can adapt to nonlinear data containing noise [43,44]. Therefore, we chose Mean Absolute Error (MAE) as our loss function, which is represented as follows:(4)$$L\left(x,x^{\prime }\right)=\frac{1}{n}\sum_{i=1}^{n}{\left(x-x^{\prime }\right)}^{2}=\frac{1}{n}\sum_{i=1}^{n}{\left(x-\sigma^{\prime }\left(W^{\prime }\left(\sigma \left(Wx+b\right)\right)\right)+b^{\prime }\right)}^{2}$$
where L is used to measure the difference between the variables x and $x^{\prime }$, which corresponds to the difference between the original input signal and the reconstructed output signal.

#### 2.2.2. TKAN

TKAN is an improved version of the Kolmogorov–Arnold Network architecture for time series, combining the classic looping and gating mechanisms of LSTM. This new architecture addresses the common long-term dependency issues found in RNNs. Compared to traditional models such as LSTM and GRU, TKAN excels in long-term prediction, demonstrating the ability to handle various situations and longer time periods [35].

The Kolmogorov–Arnold Network (KAN)’s structure is shown in Figure 3. In the presented diagram, the activation functions are not located at the nodes but are instead assigned to the connecting edges, with each edge associated with a learnable activation function parameterized as a B-spline. This configuration allows the activation functions to adapt dynamically between coarse-grained and fine-grained resolutions. Unlike traditional Multilayer Perceptrons (MLPs), which employ fixed activation functions at each node, Kolmogorov–Arnold Networks (KANs) utilize these learnable B-spline functions along the edges. The signals transmitted between nodes are aggregated using a simple summation operation, meaning that the nodes themselves only perform summation without involving activation functions. This design results in faster computational speed and improved interpretability, as the complexity of the activation functions resides in the edges rather than the nodes. The KAN network is inspired by the Kolmogorov–Arnold representation theorem [43]. The Kolmogorov–Arnold representation theorem (Equation (5)) states that any multivariate continuous function can be represented as a combination of univariate functions and addition operations:(5)$$f\left(x_{1},\ldots ,x_{n}\right)=\sum_{q=1}^{2n+1}\Phi_{q}\left(\sum_{p=1}^{n}\phi_{q,p}\left(x_{p}\right)\right)$$
where $\phi_{q,p}$ are univariate functions that map each input variable $\left(x_{p}\right)$, such $\phi_{q,p}:\left[0,1\right]\to R$ and $\phi_{q}:R\to R.$ Since all the functions to be learned are univariate, Ziming Liu et al. [34] subsequently parameterize each one-dimensional function as a B-spline curve, employing learnable coefficients for the local B-spline basis functions. So, a KAN layer is as follows:(6)$$\Phi =\{\phi_{q,p}\},\;p=1,2,\cdots ,n_{in},\;q=1,2\cdots ,n_{out},$$
where $\phi_{q,p}$ are parametrized functions of learnable parameters, and $n_{in}$ means the number of inputs, while $n_{out}$ means the number of outputs. The shape of KAN is defined as:(7)$$\left[n_{0},n_{1},\cdots ,n_{L}\right],$$
where $n_{L}$ is the number of nodes in the $i^{\mathrm{th}\;}$ layer of the computational graph. They denote the $i^{\mathrm{th}\;}$ layer by (l,i), and the activation value of the (l,i) neuron by $x_{l,i}$. Between layer l and l+1, there are $n_{l}n_{l+1}$ activation functions; the activation function that connects (l,i) and (l+1,j) is denoted by:(8)$$\phi_{l,j,i},l=0,\cdots ,L-1,i=1,\cdots ,n_{l},j=1,\cdots ,n_{l+1}.$$

The activation value of the (l+1,j) neuron is simply the sum of all incoming post-activations, and this can be written in matrix form:(9)$$x_{l+1}=\left(\begin{array}{cccc} \phi_{l,1,1}\left(\cdot \right) & \phi_{l,1,2}\left(\cdot \right) & \cdots & \phi_{l,1,n_{l}}\left(\cdot \right)\\ \phi_{l,2,1}\left(\cdot \right) & \phi_{l,2,2}\left(\cdot \right) & \cdots & \phi_{l,2,n_{l}}\left(\cdot \right)\\ \vdots & \vdots & & \vdots \\ \phi_{l,n_{l+1},1}\left(\cdot \right) & \phi_{l,n_{l+1,2}}\left(\cdot \right) & \cdots & \phi_{l,n_{l+1},n_{l}}\left(\cdot \right) \end{array}\right)x_{l},$$
where $\Phi_{l}$ is the function matrix corresponding to the $l^{th}$ KAN layer, and the notation (⋅) represents a placeholder for the input variable of a function, indicating that this position can accept a variable as an input. A general KAN network is a composition of L layers; given an input vector $x_{0}\in R^{n_{0}}$, the output of KAN is:(10)$$KAN\left(x\right)=\left(\Phi_{L-1}\circ \Phi_{L-2}\circ \cdots \circ \Phi_{1}\circ \Phi_{0}\right)x.$$
where $\Phi_{1}\circ \Phi_{0}$ = $\Phi_{1}(\Phi_{0}(x))$ represents the composition of functions.

Temporal Kolmogorov–Arnold Networks (TKAN) adapt the concept of Kolmogorov–Arnold Networks (KANs) to time series by integrating temporal management within neural networks. As shown in Figure 4, this approach combines the KAN architecture with a modified Long Short-Term Memory (LSTM) unit. The learnable activation functions in KAN capture complex nonlinear patterns in time series data, while LSTM cells, with their gating mechanisms and memory units, effectively address the challenges of long-term dependencies and the vanishing/exploding gradient problem common in traditional RNNs. This enables the model to retain information about past events over extended time horizons, enhancing its ability to model long-range temporal dependencies. Although the intricate architecture of LSTM may slightly increase computational costs, this trade-off is justified in specific learning tasks such as pipeline leak detection, where high accuracy is paramount.

We denote the input vector of dimension *d* as $x_{t}$. This unit uses several internal vectors and gates to manage information flow. The forget gate, with activation vector $f_{t}$, shown as follows:(11)$$f_{t}=\sigma \left(W_{f}x_{t}+U_{f}h_{t-1}+b_{f}\right),$$
decides what information to forget from the previous state. The input gate, with the activation vector denoted as $i_{t}$, shown as follows:(12)$$i_{t}=\sigma \left(W_{i}x_{t}+U_{i}h_{t-1}+b_{i}\right),$$
controls which new information to include.

In the above, σ represents the activation function, $x_{t}\;$denotes the input vector at the current time step, $h_{t-1}\;$represents the hidden state vector from the previous time step, W is the weight matrix connecting the input vector, U is the weight matrix connecting the hidden state vector from the previous time step $h_{t-1}$, and $b_{i}\;$denotes the bias vector.

The KAN network now embeds memory management at each layer:(13)$$KAN\left(x,t\right)=\left(\Phi_{L-1,t}\circ \Phi_{L-2,t}\circ \cdots \circ \Phi_{1,t}\circ \Phi_{0,t}\right)\left(x,t\right).$$

The output gate, with activation vector $o_{t}$, shown as follows:(14)$$o_{t}=\sigma \left(KAN\left(\vec{x},t\right)\right),$$
determines what information from the current state to output given $KAN\left(\vec{x},t\right)$. The hidden state, $h_{t}$, captures the unit’s output, while the cell state, $c_{t}$ is updated as follows:(15)$$c_{t}=f_{t}⊙c_{t-1}+i_{t}⊙{\tilde{c}}_{t},$$
where ⊙ represents element-wise multiplication (the Hadamard product), and ${\tilde{c}}_{t}=\sigma \left(W_{c}x_{t}+U_{c}h_{t-1}+b_{c}\right)$ represents its internal memory. All these internal states have a dimensionality of h. The ouput denoted $h_{t}$ is given by:(16)$$h_{t}=o_{t}⊙tanh\left(c_{t}\right).$$

#### 2.2.3. Proposed Anomaly Detection TKAN-AE Model

In this study, we focus on the negative pressure wave signals generated by pipeline leaks, which are time series data. Therefore, we define this problem as a time series anomaly detection task. Time series anomaly detection aims to identify and detect abnormal patterns or behaviors in time series data, which may indicate errors, unexpected events, or system failures [45].

To address the specific requirements of time series anomaly detection, we developed a novel model named TKAN-AE (as illustrated in Figure 5, showing the detailed structure of the proposed TKAN-AE network). This model integrates the Temporal Kolmogorov–Arnold Network (TKAN) within an autoencoder (AE) architecture, effectively capturing salient features of time series data. The TKAN component combines the temporal gating mechanism of Long Short-Term Memory (LSTM) networks with the powerful function approximation capabilities of Kolmogorov–Arnold Networks (KANs), thus capturing both long-term and short-term dependencies inherent in time series data. Meanwhile, the autoencoder’s encoder–decoder structure facilitates the learning of efficient data representations, reconstructing the input to identify anomalies.

In particular, during the encoding phase, the TKAN layer initially processes the input time series data. The primary objective of this layer is to maintain the structural coherence of the time series, thereby enabling in-depth analysis of temporal dependencies. The TKAN preserves the output of each timestep, allowing the model to effectively process temporal information in subsequent layers and capture long-term relationships across the sequence. Subsequent encoding layers aggregate the temporal information into a compact fixed-size feature vector, which serves as the basis for reconstructing the original time series during the decoding phase.

In the decoding phase, the TKAN layer’s role is to restore the fixed-size feature vector, derived during encoding, back to the original time series format. By expanding each element of the feature vector temporally, this layer reconstructs the output sequence, ensuring it matches the length of the original input. The reconstruction process occurs progressively, as each timestep is reconstructed using information encoded within the learned feature representation. By reconstructing each timestep with precision, the TKAN layer enables the model to effectively recover the complete time series from its compressed representation, which is critical for accurate reconstruction and the identification of anomalies.

### 2.3. A Novel Anomaly Detection and Sequence Labeling Method for Pipeline Leak Detection

Reconstruction error is defined as the difference between the reconstructed data and the original data. In anomaly detection methods based on reconstruction error, it is generally assumed that the reconstruction error of normal data is small, while the reconstruction error of anomalous data is large. As illustrated in Figure 6 when a pipeline leak occurs (the bottom subplot in Figure 6), the reconstruction error (or loss) typically increases sharply in the initial phase (the upper subplot in Figure 6), but it may gradually decrease as the leak continues. This phenomenon arises because the model gradually learns the characteristics of the leakage process and attempts to reconstruct and predict it. As the negative pressure wave propagates through the pipeline system, the internal pressure of the pipeline rises significantly. For a model that has already learned the anomalous features, this sudden pressure increase leads to a sharp rise in the reconstruction error. However, in traditional threshold-based detection methods, data are classified as anomalous only when the reconstruction error exceeds a predefined threshold. Due to the decrease in the reconstruction error during the leakage process, certain signals that should be identified as indicative of a leak may be overlooked in specific local regions, resulting in missed detections, as shown in the yellow box. Furthermore, the subsequent rise in pressure caused by the propagation of the negative pressure wave can lead to an increase in the reconstruction error, thereby causing false positives, as shown in the green box. The issue of threshold selection has long been a challenge in anomaly detection. Research by Xue Yang et al. [46] has shown that fixed or static thresholds often fail to adapt to the dynamic characteristics of real-world data, resulting in false positives or missed detections. As shown in Figure 7, setting thresholds too low (Figure 7a) or too high (Figure 7b) can lead to significant false positives or false negatives, which, in industrial applications, may result in serious losses and accidents.

To address this issue, we propose an improved approach that builds upon the traditional reconstruction error framework. By conducting a detailed analysis of the reconstruction error curve and incorporating prior knowledge along with the model’s core principles, the goal is to enhance the performance of anomaly detection. The specific method is as follows.

As illustrated in Figure 8, showing the flowchart of anomaly detection and labeling methods for pipeline leaks, the specific detection procedure is as follows: We initially identify point P1, corresponding to the maximum reconstruction error. Subsequently, within the sequence leading up to P1, we determine point P2, which represents the minimum loss value. Finally, we locate point P3, characterized by the maximum reconstruction error preceding P2. The sequence between points P1 and P3 is defined as the potentially anomalous region, which is highly indicative of leakage. The detailed experimental comparison process and results will be elaborated in the subsequent sections.

## 3. Benchmark Dataset

In this section, we constructed a benchmark dataset based on pressure signals collected from field experiments. Pressure signals, recognized as a convenient and stable indicator for pipeline leakage detection [46,47,48,49], are widely employed in leakage detection tasks. To ensure the practical relevance of our study, we deliberately avoided using simulated data generated by software such as Fluent or CFX, as well as data from laboratory-based pipeline systems. Instead, we utilized field leakage experiment data from actual urban water supply pipelines, as these data more accurately reflect real-world industrial applications. In contrast, simulated data often exhibit biases that are more aligned with specific experimental conditions, lacking the broader applicability to real-world usage scenarios.

The field experiments were conducted on water supply pipelines located underground at Shenjiang Road and Wuzhou Avenue in Pudong New District, Shanghai, China (the yellow line in Figure 9a). The total length of the pipeline involved in the experiment was 5891 m. High-frequency dynamic pressure sensors were installed at wells J15 and J23 (Figure 9b) to monitor pressure fluctuations. To simulate leak events, a valve located at well J22 was manipulated to induce leakage, with different valve openings representing varying leakage severities (Figure 10a), and the sensor installation diagram for the field site is shown in Figure 10b. Furthermore, the exact times for opening and closing the valves were recorded to provide reference data for subsequent model evaluation. The distance between J15 and J22 was approximately 4981 m, that between J15 and J23 was about 5891 m, and that between J22 and J23 was around 910 m.

To ensure the realistic simulation of leakage scenarios and eliminate randomness, the experiments were conducted over multiple days and at different times of day, with sensor data continuously collected over an extended period, thereby ensuring data diversity and representativeness. An overview of the leakage data is provided in Table 1. Additionally, pressure signals under normal operating conditions were collected over several days to serve as reference data for healthy states. Based on these experiments, we found that the number of normal samples greatly exceeded the number of leakage samples. To address the potential imbalance in the dataset, which could affect model training, we ultimately selected 12 leakage samples and 200 normal samples to construct the dataset. This balanced selection of data aimed to ensure that the model could effectively learn from the limited leakage data while preventing bias caused by the disproportionate number of normal samples. The raw pressure signals were processed using the data preprocessing methods outlined in Section 2.1. These signals were then segmented into subsequences of equal length. After extensive experimentation, we selected a standard subsequence length of 13,000 data points to create our benchmark dataset. A shorter sequence would have failed to capture the complete leak process, while a longer sequence might have included pressure fluctuations associated with multiple states, which would have reduced the data consistency and hindered the learning process of the model. Moreover, for time series models, excessively long sequences significantly increase the computational load and memory requirements. Thus, a sequence length of 13,000 data points was a suitable choice given our current computational capacity, striking a balance between computational efficiency and data representativeness. This dataset was subsequently used to train our model and to compare its performance with other benchmark models. An overview of the benchmark test dataset is presented in Table 2.

Figure 11 provides an example of pressure signal data, illustrating that a sharp drop in pressure occurs when a pipeline experiences a leakage event. This phenomenon is primarily caused by the rapid outflow of fluid at the leakage point, resulting in a localized pressure reduction. Subsequently, due to the generation of a negative pressure wave and its propagation towards both ends of the pipeline, significant changes occur in the internal pressure dynamics of the pipeline. The propagation of the negative pressure wave induces fluid near the leakage point to move towards the low-pressure zone, leading to a transient pressure rise and forming a U-shaped waveform. Over time, as the negative pressure wave reflects and attenuates within the pipeline, the system eventually reaches a new steady-state pressure, which, in the short term, falls below the original operating pressure. This temporary reduction in steady-state pressure is due to the ongoing fluid loss at the leakage point.

## 4. Experiment Results and Analysis

### 4.1. Experiment Environment

Our model construction and experimental analyses were performed on Google Colaboratory (Colab). All experiments utilized a Tesla T4 GPU manufactured by NVIDIA Corporation in Santa Clara, California, United States, accessed through Google Colab, equipped with CUDA Version 12.2 and an Intel(R) Xeon(R) CPU manufactured by Intel Corporation in Santa Clara, CA, USA, @ 2.20 GHz, paired with 12 GB of RAM. The software version details are as follows:-Python: 3.10.12-TensorFlow: 2.15.0-Keras: 2.15.0

Google Colaboratory is a cloud-based, interactive notebook environment developed by Google. It provides users with free computational resources, including CPUs, GPUs, and TPUs, allowing them to execute code directly in a web browser without requiring any local setup or installation. TensorFlow, an open-source machine learning framework, was used to facilitate the construction, training, evaluation, and saving of the neural network models. Keras, integrated within TensorFlow, is a high-level neural network API that provides models, thereby simplifying the development of deep learning workflows in Python.

### 4.2. Metrics for Model Performance

Several evaluation metrics are commonly used to assess the performance of classification machine learning models. For pipeline leakage time series anomaly detection, the following metrics are particularly suitable:

Accuracy: This is a fundamental metric that measures the proportion of correctly predicted instances out of the total number of instances. In general classification tasks, for the time series anomaly detection of pipeline leaks, a continuous leak anomaly segment is regarded as a whole (a positive sample). If the detector triggers an alarm at any point within the segment and identifies an anomaly, then the entire segment as a sample is considered successfully detected. However, evaluating the entirety of the sample does not adequately measure the performance of the detector, and subsequent work on leak localization heavily relies on the precision of detection. Therefore, we introduce a new metric—segment-wise precision [50,51,52,53].

Segment-wise precision: This metric measures the proportion of actual anomaly points within a specific segment that has been detected as anomalous. In other words, after a sample is successfully detected as a whole, we continue to focus on how many points within it were correctly identified as anomalies.

True positives (TPs): The number of actual leakage instances correctly identified by the model as leaks.

False negatives (FNs): The number of actual leakage instances that the model failed to identify as leaks.

False positives (FPs): The number of non-leakage instances incorrectly classified by the model as leaks.

True negatives (TNs): The number of non-leakage instances correctly identified by the model as non-leaks.

These metrics are often visualized using a confusion matrix, which provides a comprehensive overview of the model’s classification performance.

Confusion matrix: A tool used to evaluate the performance of classification models by comparing the model’s predictions to the actual outcomes. It includes four components: true positives (TPs), false positives (FPs), true negatives (TNs), and false negatives (FNs). The confusion matrix helps to visualize the model’s accuracy, false-positive rate, and false-negative rate, thus providing a holistic evaluation of its performance.

Precision: Precision measures the proportion of true-positive predictions among all positive predictions made by the model. It reflects the reliability of the model’s positive predictions.

Recall: Recall measures the proportion of actual positive instances that are correctly identified by the model. It reflects the model’s ability to identify all relevant instances.

F1-score: The F1-score is the harmonic mean of precision and recall, providing a balanced evaluation of the model’s classification performance, particularly in cases where there is an uneven class distribution.

These metrics are critical for a comprehensive assessment of pipeline leakage detection models in time series anomaly detection tasks. The mathematical expressions for these metrics are provided in Equations (17)–(21).(17)$$\mathrm{Accuracy}=\frac{TP}{TP+FP+TN+FN}$$(18)$$Segment-wise\;Precision=\frac{\left|S_{detected}\cap S_{true}\right|}{\left|S_{detected}\right|}$$

$S_{detected}$ represents the set of detected anomalous segments, while $S_{true}$ represents the set of actual anomalous segments.(19)$$Precision=\frac{TP}{TP+FP}$$(20)$$Recall=\frac{TP}{TP+FN}$$(21)$$F1-\mathrm{score}=2\times \frac{\mathrm{Precision}\times \mathrm{Recall}}{\mathrm{Precision}+\mathrm{Recall}}$$

### 4.3. Experiment Development

Building on the work of Ziming Liu et al. [34], which demonstrated that KAN networks effectively fit data using a minimal number of parameters and shallow network architectures, we applied similar principles to develop our TKAN-AE network. Both the encoder and decoder in our TKAN-AE network consist of a single TKAN layer, utilizing B-spline activation functions of orders ranging from 0 to 4. For benchmarking, we selected two widely used recurrent neural network (RNN) models for multistep prediction: the Gated Recurrent Unit (GRU) and the Long Short-Term Memory network (LSTM). Although the Transformer architecture has the following disadvantages due to its extensive parameter requirements, primarily resulting from the multi-headed attention mechanism and numerous fully connected layers, which render it unsuitable for low-cost, large-scale deployment in industrial applications, we still built and conducted relevant comparison experiments to further validate the generalization capability of the proposed labeling method across different models. Ultimately, we selected a basic Transformer model and an Informer model specifically developed for time series. Informer is a Transformer-based model designed for long-sequence time series forecasting. By introducing a sparse attention mechanism and optimizing the architectural design, Informer enhances computational efficiency and prediction accuracy. To ensure a fair comparison, we also constructed simplified autoencoder variants, namely, LSTM-AE, GRU-AE, Transformer–AE, and Informer–AE, each using a single-layer LSTM, GRU, Transformer, or Informer architecture for both the encoder and decoder components. All models were trained using the Adam optimizer, an adaptive moment estimation method introduced by Kingma et al. [42], which combines the benefits of the momentum and RMSProp algorithms. The loss function for all models was the Mean Absolute Error (MAE). To mitigate overfitting, dropout regularization was applied with a probability of 0.2, alongside a weight decay of 0.2, following recommendations by Kumar and Asri et al. [46,47]. Furthermore, to further compare the proposed anomalous sequence labeling method with traditional threshold-based approaches, after experimenting with various thresholds and analyzing the kernel density estimation (KDE) plot of reconstruction errors, the threshold that achieved the best detection performance on the validation set—such as the lowest false-positive rate and false-negative rate—was ultimately selected. This study determined 1 to be the optimal threshold. The detailed steps and implementation process for this selection, including the corresponding code, are provided in the Data Availability Statement, where the open-source GitHub repository contains executable Colab code.

In line with the findings of Aburass et al. [50], we recognize that an excessive increase in the number of training epochs does not necessarily improve detection accuracy and may instead lead to higher computational costs. Their research highlights diminishing returns in performance as the number of epochs increases, particularly for models designed to detect anomalies in time series data, rather than solely focusing on model fitting. Therefore, we tested multiple training epoch configurations to determine the optimal number of epochs. As shown in Figure 12, the best performance was achieved at 30 epochs, but the improvement over 20 epochs was marginal, while computational costs increased by 33%. Based on this, we selected 20 epochs as the standard training epoch count for all subsequent comparative experiments.

### 4.4. Results and Comparison

In this experiment, we compared the performance of six detectors, resulting from the combination of five models (TKAN-AE, LSTM-AE, GRU-AE, Transformer–AE, and Informer–AE) with two different anomalous sequence labeling methods (Figure 13). We have provided links to publicly available runnable code for the comparative example experiments in the Data Availability Statement section, allowing others to conduct reproducible experiments and further develop in-depth research based on our work. This comparison not only evaluated the performance of each model but also verified the superiority and robustness of the proposed labeling approach. Figure 14 presents specific detection instances, where subfigures (a), (b), (c), (d), and (e), respectively, illustrate the visual comparison between our proposed method and the traditional methods for the five models. Through the study of specific leakage detection tasks, we observed that for the same model, the differences brought by the different labeling methods were quite significant. The traditional labeling method based on reconstruction error thresholds tended to miss a substantial number of intermediate leakage segments, and it also incorrectly labeled a short sequence as anomalous after the leakage had ended, leading to notable false negatives and false positives (Figure 14). In contrast, our proposed new labeling method effectively avoided these issues, significantly improving the labeling accuracy. Furthermore, under the same labeling method, the detection capabilities of the five models weare generally consistent. Figure 15 further provides a detailed comparison of the segment-wise precision between the two labeling approaches. Our method outperformed the traditional methods by 54.2% in terms of segment-wise precision.

From the detection instances (Figure 14), it is evident that all ten detectors performed well and met the requirements for industrial applications. Notably, TKAN-AE, along with the other four established models, maintained strong performance even when there were abrupt changes in the original data, demonstrating the models’ robustness in complex environments. Additionally, our proposed labeling method was successfully applied across different models, providing strong experimental support for its generalization capabilities, while also reducing the false negatives and false positives commonly seen in traditional methods.

Table 3 presents a comparative analysis of the aforementioned five models, emphasizing fairness by employing the newly proposed annotation method across all models. By examining the confusion matrices of the five models in Figure 16a–e, it is evident that all models demonstrated excellent anomaly detection accuracy and the ability to identify specific abnormal sequences on the actual dataset. TKAN, Transformer, and Informer exhibited balanced performance, maintaining good consistency between precision and recall and achieving high F1 scores, which highlights the overall stability of these models. In contrast, LSTM showed a higher recall but a lower precision compared to TKAN, indicating a tendency to produce more false positives. Additionally, the relatively lower recall of GRU suggests potential issues with missing certain anomalies, revealing its limitations.

In terms of memory usage (measured by the total number of parameters), the autoencoder (AE) networks based on the TKAN and GRU architectures have fewer parameters, whereas the Transformer and Informer models possess a significantly larger number of parameters, far exceeding that of TKAN. This makes them difficult to deploy on low-cost devices. Regarding training speed on GPUs, the traditional time series models significantly outperformed the TKAN network. However, the training speed of Transformer and Informer was relatively slow, although Informer performed slightly better due to its structural adjustments specifically tailored for time series data. It is important to note that as a recently proposed model, the KAN network has not yet been optimized for efficiency or specifically adapted to GPU architectures, resulting in relatively poor GPU compatibility. Additionally, from a technical perspective, the introduction of learnable activation functions enhances the model’s interpretability, but compared to fixed activation functions, it incurs higher evaluation costs, thereby slowing down the training speed. This issue has also been emphasized by Liu Ziming, the author of the KAN model. Interestingly, when training on CPUs, the TKAN network surpassed the traditional LSTM models in terms of training speed, which may have been due to the CPU architecture being more suitable for handling complex sequence computations. Notably, the TKAN model even outperformed the GRU model, which is renowned for its lightweight and fast training capabilities. On the other hand, the training time for Transformer and Informer increased significantly, especially as the training time for a single sequence using the standard Transformer model reached the level of 20 min. This renders them completely unsuitable for direct application on industrial-grade devices for real-time leak detection tasks in the field.

## 5. Discussion

Although the proposed method introduces a novel approach for RNN-based models, which traditionally rely heavily on MLP architectures, and although it demonstrated strong performance on the real-world pipeline datasets, several limitations remain:

1. Data Scarcity: Despite utilizing a real-world pipeline dataset rarely employed in previous studies, the scarcity of anomaly samples presents a significant challenge, largely due to the high costs and safety considerations associated with leakage experiments. Moreover, the completely unsupervised learning approach employed in this study did not fully maximize the use of these rare anomaly samples, which continues to be a challenge in the field of anomaly detection. Potential solutions include:

(1) Transfer Learning: By combining laboratory pipeline datasets with real-world datasets, transfer learning can effectively leverage existing knowledge to enhance model performance in new environments at reduced costs.

(2) Semi-Supervised Learning: As demonstrated by Guansong Pang et al. [51], semi-supervised learning is one of the most effective methods in anomaly detection. This approach combines a small amount of labeled anomaly data with a large volume of unlabeled normal data for training. By learning the characteristics of normal data, semi-supervised learning can more accurately identify anomalies, addressing the inefficiency of unsupervised learning in utilizing known anomaly information while also avoiding the reliance on large amounts of labeled data typical of fully supervised methods [51,52,53].

2. Model Adaptability: The proposed model is tailored specifically for anomaly detection in a particular urban pipeline network. Unlike fluid-based models, our method benefits from not requiring knowledge of physical parameters such as pipeline geometry, fluid properties, or pressure, instead relying directly on monitoring data to train the leakage detection model. These monitoring data inherently contain physical information about the pipeline network. However, if the configuration of the pipeline network changes, the model must be retrained. Recent advances in federated learning have shown promise in addressing this adaptability issue. Federated learning integrates resources and updates models in real time, while reducing computational costs through distributed computing. More importantly, in an era where data security is of paramount concern, federated learning’s robust data privacy protection capabilities have gained considerable attention.

3. Model Interpretability: The model proposed in this paper leverages the powerful fitting capability and relatively low parameter complexity of the KAN network. Additionally, the KAN network possesses an important feature: interpretability. This feature is crucial for pipeline leak point localization. The negative pressure wave time difference positioning method is widely used to identify pipeline leaks. This method utilizes the time differences in negative pressure waves arriving at multiple sensors along the pipeline and calculates the leak location based on the wave speed of the negative pressure waves. Typically, the negative pressure wave speed is determined either through empirical formulas or through complex numerical simulations involving partial differential equations (PDEs). Enhancing the interpretability of the KAN network can improve the visualization of the PDE solving process, thereby increasing the speed and accuracy of leak localization. Therefore, using the interpretability of the KAN network for leak point localization is an important direction for future research.

## 6. Conclusions

1. The proposed TKAN-AE model demonstrates exceptional performance and stability, effectively identifying signal features associated with pipeline leaks. Compared to traditional time series models, TKAN-AE significantly reduces the number of parameters, making it particularly suitable for large-scale, cost-efficient industrial deployment. Furthermore, this model offers a novel alternative to traditional fully connected networks, such as MLP.

2. The improved anomaly sequence labeling method substantially enhances the accuracy of time series anomaly detection, achieving a segment-level detection precision of 93.1% for specific anomalous sequences. The application of this method across five different models further confirms its strong generalization capability and robustness.

3. This study validates the proposed detector using real-world pipeline leakage experimental data from a megacity. The results demonstrate superior performance and robustness, providing strong support for the methods potential application in industrial settings.

## Figures and Tables

**Figure 1 sensors-25-00384-f001:**
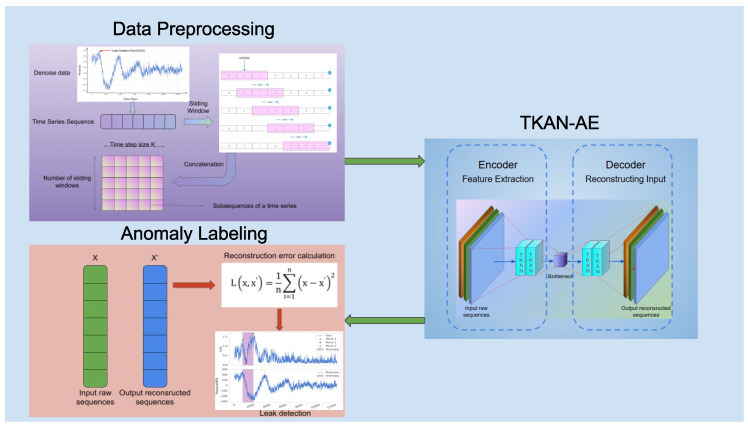
The overall structure of the detector.

**Figure 2 sensors-25-00384-f002:**
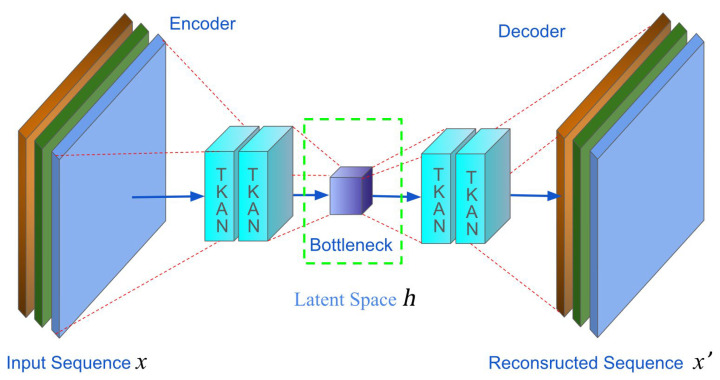
The structure of an autoencoder.

**Figure 3 sensors-25-00384-f003:**
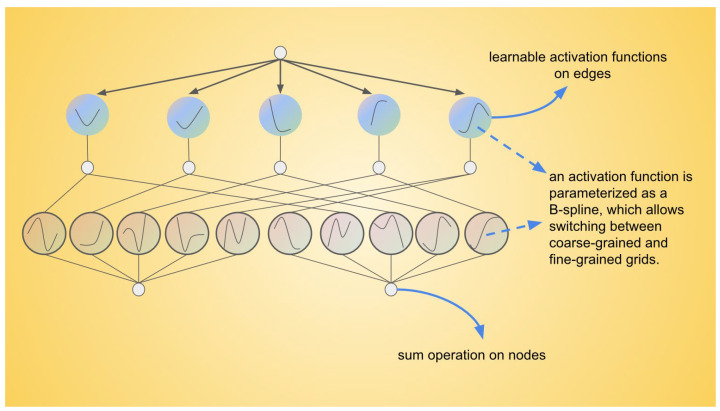
The detailed structure of the KAN network.

**Figure 4 sensors-25-00384-f004:**
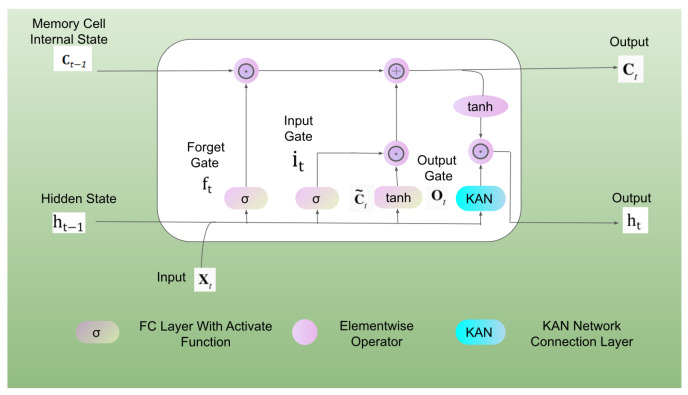
The detailed structure of the TKAN network.

**Figure 5 sensors-25-00384-f005:**
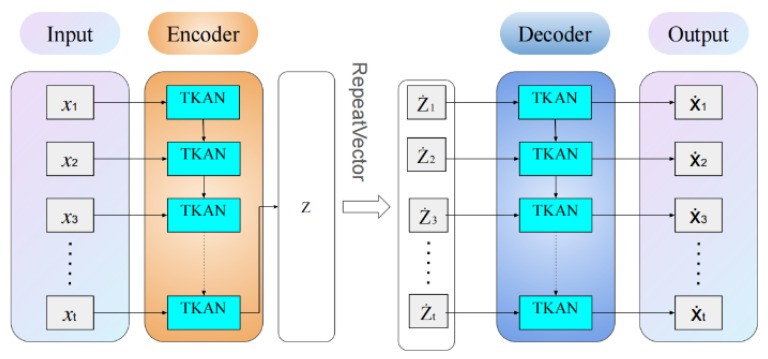
The detailed structure of the proposed TKAN-AE network.

**Figure 6 sensors-25-00384-f006:**
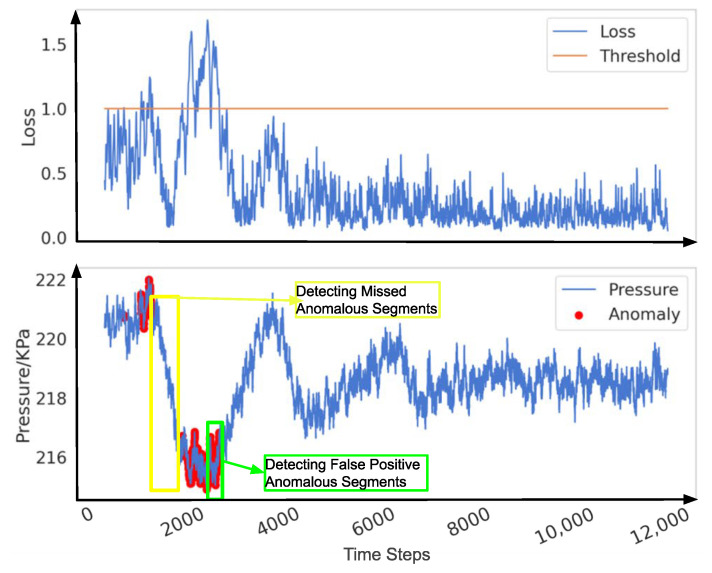
The issues in traditional threshold-based anomaly detection methods.

**Figure 7 sensors-25-00384-f007:**
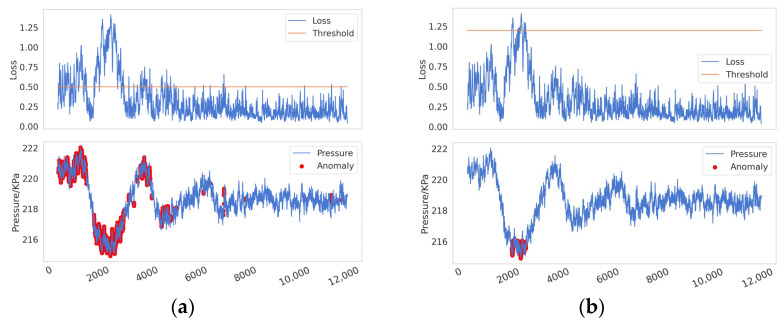
The problems caused by setting thresholds too low (**a**) or too high (**b**).

**Figure 8 sensors-25-00384-f008:**
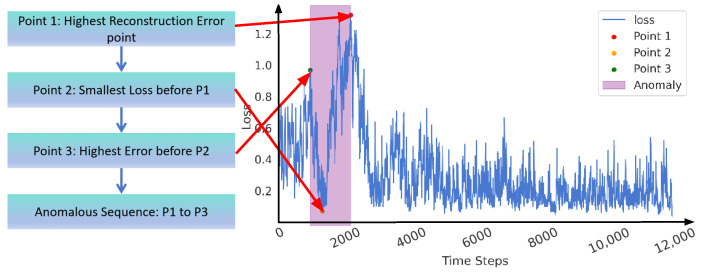
The flowchart of anomaly detection and labeling methods for pipeline leaks.

**Figure 9 sensors-25-00384-f009:**
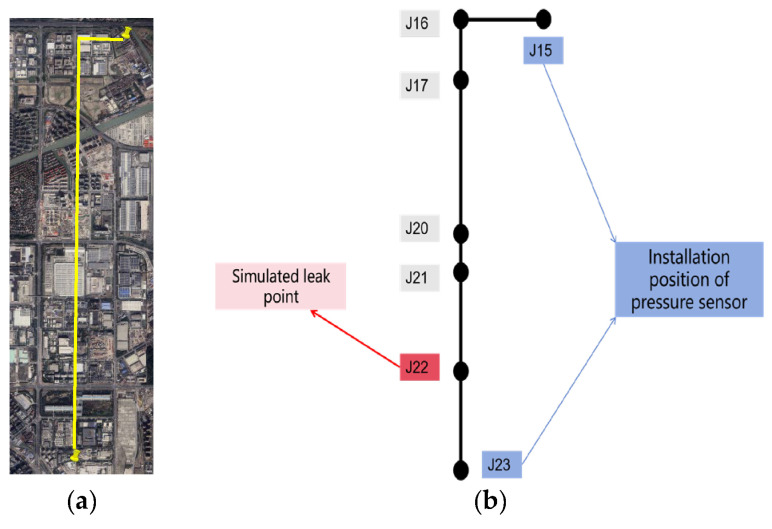
An overview of the pipeline from a satellite map and a topographic map perspective: (**a**) overview map of pipelines (real urban water supply pipelines in Shanghai); (**b**) layout map of pipelines (including sensors and simulated leakage locations).

**Figure 10 sensors-25-00384-f010:**
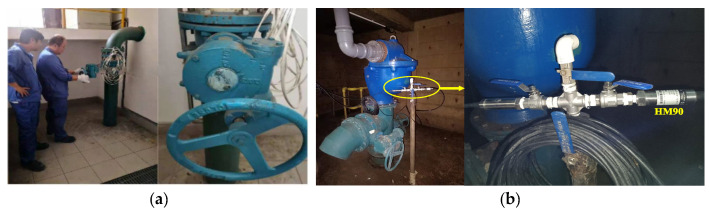
On-site experiment illustrations: (**a**) workers’ operations with varying valve openings during leak simulation; (**b**) pressure sensor installation.

**Figure 11 sensors-25-00384-f011:**
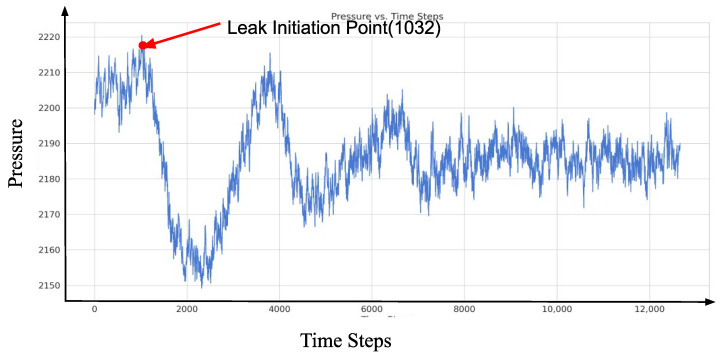
A sample of pressure signal data.

**Figure 12 sensors-25-00384-f012:**
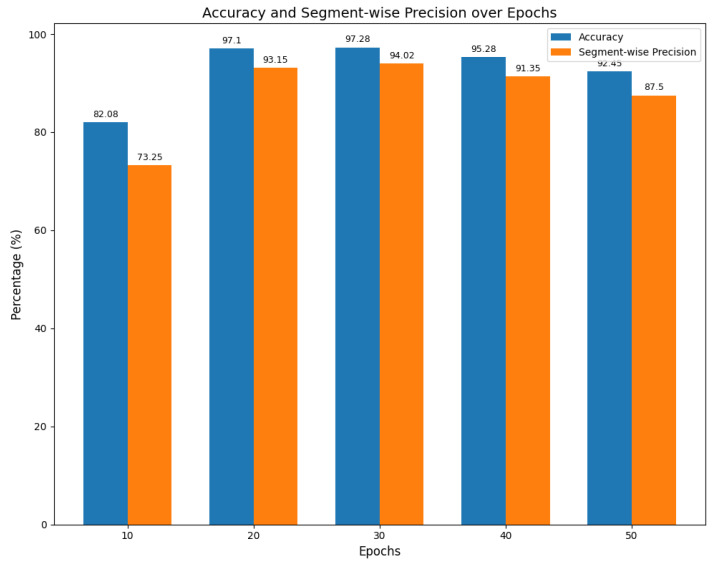
Accuracy and segment-wise precision over epochs.

**Figure 13 sensors-25-00384-f013:**
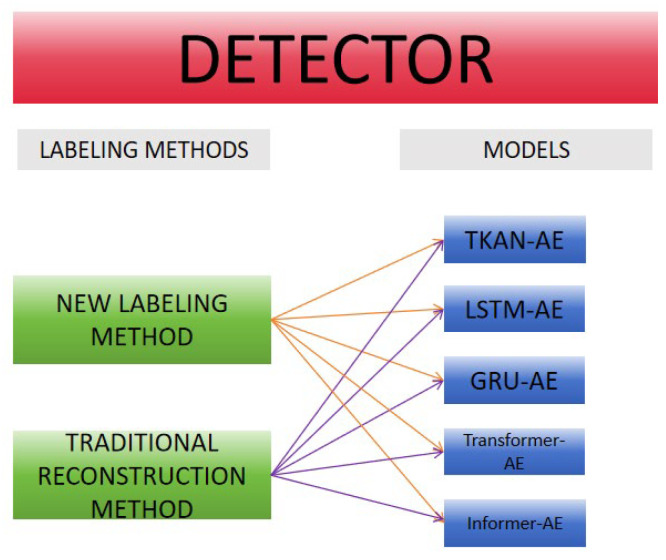
The composition of the six detectors in the comparative experiment.

**Figure 14 sensors-25-00384-f014:**
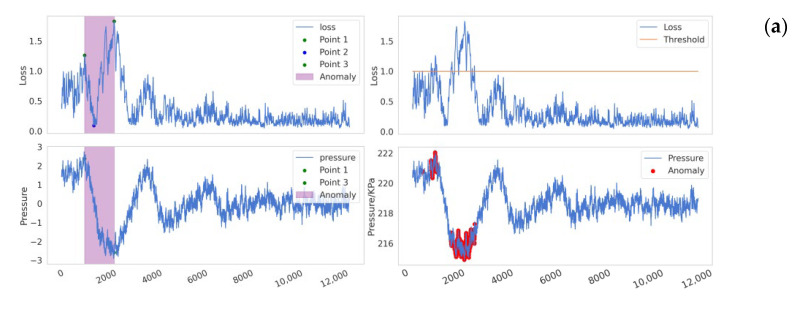
The performance of ten detection methods on leakage sample segments. The figures labeled (**a**–**e**) respectively present detection instance demonstrations for five models—TKAN, LSTM, GRU, Transformer, and Informer—each combined with an Autoencoder, evaluated under two different labeling methods.

**Figure 15 sensors-25-00384-f015:**
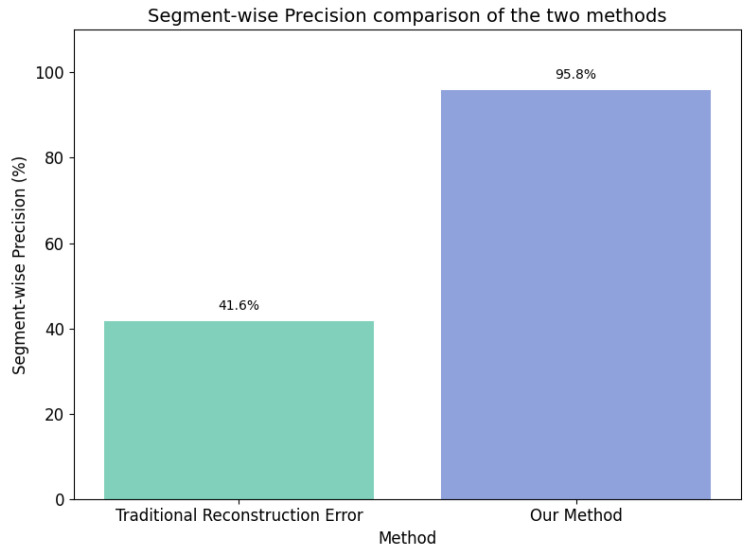
Segment-wise precision of two methods for specific detection instances.

**Figure 16 sensors-25-00384-f016:**
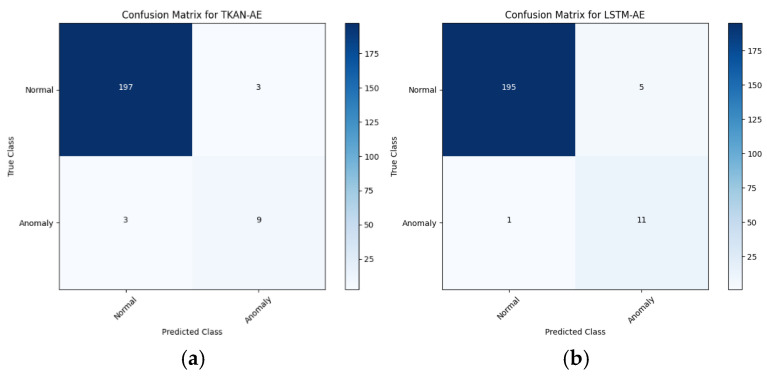
The confusion matrices of five models. Subfigures (**a**)–(**e**) respectively illustrate the detection confusion matrices for five models—TKAN, LSTM, GRU, Transformer, and Informer—each integrated with an Autoencoder and all employing our proposed anomalous sequence labeling approach.

**Table 1 sensors-25-00384-t001:** Properties of the leakage experiment data.

Dataset Properties	Value
Number of healthy samples	200
Number of samples with leakage	12
The signal length of a single sample	13,000-time steps
Total signal length ^1^	2,756,000-time steps
Resolution: sampling frequency (Hz)	500
Anomaly ratio ^2^	5.66%

^1^ Total signal length = the signal length of a single sample × total samples = 13,000 × 212 = 2,756,000. ^2^ Anomaly ratio: the proportion of samples labeled as anomaly points to the total number of samples in the dataset. This metric was used to quantify the rarity or prevalence of anomalies within the dataset. Anomaly ratio = 12/212 = 5.66%.

**Table 2 sensors-25-00384-t002:** Benchmark dataset properties acquired from the experimental field tests.

Valve Opening Degree	Number of Experiments	Simulated Leakage Area S_L_ (dm^2^)	Pipe Cross-Sectional Area S_P_ (dm^2^)	S_L_/S_P_ ^1^
30	3	1.05	1017.9	0.103%
45	3	1.57	1017.9	0.154%
60	3	2.36	1017.9	0.232%
90	3	3.14	1017.9	0.308%

^1^ S_L_/S_P_: the primary function of this indicator is to reflect the proportion of the leakage area relative to the cross-sectional area of the pipe, quantifying the severity of the leakage.

**Table 3 sensors-25-00384-t003:** Performance comparison between the proposed networks and benchmark time series models across eight metrics.

Model	Accuracy (%)	Segment-Wise Precision (%)	Precision (%)	Recall (%)	F1-Score (%)	Total Parameters ^1^	GPU Cost (s)	CPU Cost (s)
TKAN-AE	97.1	93.1	75.0	75.0	75.0	73,544	188.9	125.8
LSTM-AE	98.1	93.0	68.7	91.6	78.3	96,331	23.2	162.0
GRU-AE	96.6	92.8	72.7	66.7	69.5	73,631	12.8	150.7
Transformer–AE	97.1	94.2	75.0	75.0	75.0	129,131	138.3	1337.7
Informer–AE	98.1	83.3	83.3	83.3	83.3	503,641	51.1	140.9

^1^ Total Parameters: The total parameter count represents the sum of all trainable parameters in a deep learning model, serving as an indicator of the model’s complexity. A larger number of parameters generally enhances the model’s learning capacity, enabling it to capture more intricate patterns and details. However, an excessive number of parameters may lead to overfitting, particularly when the dataset is insufficient. Additionally, larger models require greater computational resources, resulting in higher training and inference costs. Therefore, the total parameter count is a crucial metric for evaluating both the complexity of the model and the associated computational resource demands.

## Data Availability

The code used as demonstration in this paper has now been open-sourced on GitHub. It can be run directly on Colab to reproduce the experimental results presented in this work. The link to access it is: https://github.com/wuhyyy/TKAN-AE. Due to the confidential nature of the field test data, I can only provide the data segments used in the sample detection example presented in the paper. You may access them via the following Google Drive link. https://drive.google.com/file/d/1zdJppw9UAEPFWIKY9jaqlAGUazse8Zyc/view?usp=sharing.

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
