# Peer review of "Research on a Novel Unsupervised-Learning-Based Pipeline Leak Detection Method Based on Temporal Kolmogorov–Arnold Network with Autoencoder Integration"

_sensors, 2025, doi:10.3390/s25020384_

Round 1

Reviewer 1 Report

Comments and Suggestions for Authors

1. The author said that the innovation of this paper is to propose a hybrid model called TKAN-Autoencoder (TKAN-AE). But why the hybrid model is better than the single model is not well described in the introduction. Therefore, the title, abstract and introduction of this paper should be further modified to highlight the innovation of this paper.

2. In this study, the time domain characteristics of pressure signals are used to predict pipeline leakage, so how much resolution can be achieved by this method? Whether other types of signals are collected, such as vibration signals or acoustic emission signals.

3. It seems that the threshold value is not a certain value in Fig.13, so how to determine the threshold value according to the actual operating conditions?

4.In the comparison of methods Fig.14, why only 41.6% of the methods are compared, and is it reasonable?

5. Fig.6, 7 and 10 are lack of labels or units for horizontal or vertical coordinates.

Comments on the Quality of English Language

The English could be improved to more clearly express the research.

Author Response

请参阅附件。

Reviewer 2 Report

Comments and Suggestions for Authors

This paper presents a novel model (TKAN-AE) combining Temporal Kolmogorov-Arnold Network (TKAN) with Autoencoder (AE) for pipeline leakage detection. However, here are some issues that should be considered.

(1) The paper proposes a novel anomalous sequence labeling method based on reconstructed error curves and a priori knowledge, which can capture anomalies in the leakage process more accurately than the traditional thresholding method. It is recommended to detail the specific implementation steps of this method and explain its advantages over traditional methods, especially when dealing with complex time series data.

(2) In addition to comparing with traditional RNN models such as LSTM, GRU, etc., it is recommended to consider comparing with other cutting-edge time series anomaly detection methods (e.g., Transformer-based models, graph neural networks, etc.) in order to emphasize the innovation and superiority of the TKAN-AE model.

(3) The paper mentions that the KAN network has strong explanatory properties, but the current experimental results mainly focus on the performance of the model. Please further explore how the explanatory properties of KAN networks can be exploited, especially in the task of leakage point localization, and how the localization accuracy can be improved by analyzing physical features such as the propagation speed of negative pressure waves?.

Author Response

请参阅附件。

Round 2

Reviewer 1 Report

Comments and Suggestions for Authors

The author has revised the manuscript according to the comments of the last review, and it is recommended for acceptance now.